# Deep Eutectic Solvents: Alternative Solvents for Biomass-Based Waste Valorization

**DOI:** 10.3390/molecules27196606

**Published:** 2022-10-05

**Authors:** Giovanni P. Rachiero, Paula Berton, Julia Shamshina

**Affiliations:** 1Valagro S.p.A., Via Cagliari 1, 66041 Atessa, Chieti, Italy; 2Chemical and Petroleum Engineering Department, University of Calgary, Calgary, AB T2N 1N4, Canada; 3Fiber and Biopolymer Research Institute (FBRI), Department of Plant and Soil Science, Texas Tech University, Lubbock, TX 79409, USA

**Keywords:** biomass recovery, deep eutectic solvents, ionic liquids, circular economy

## Abstract

Innovative technologies can transform what are now considered “waste streams” into feedstocks for a range of products. Indeed, the use of biomass as a source of biopolymers and chemicals currently has a consolidated economic dimension, with well-developed and regulated markets, in which the evaluation of the manufacturing processes relies on specific criteria such as purity and yield, and respects defined regulatory parameters for the process safety. In this context, ionic liquids and deep eutectic solvents have been proposed as environmentally friendly solvents for applications related to biomass waste valorization. This mini-review draws attention to some recent advancements in the use of a series of new-solvent technologies, with an emphasis on deep eutectic solvents (DESs) as key players in the development of new processes for biomass waste valorization. This work aims to highlight the role and importance of DESs in the following three strategic areas: chitin recovery from biomass and isolation of valuable chemicals and biofuels from biomass waste streams.

## 1. Introduction

Biomass waste streams are potential feedstocks for a variety of products such as fuels, polymers, and building blocks. Nowadays, most biomass residues are either not utilized or significantly under-utilized, i.e., left in the field to decompose naturally, be openly burned, or be used as a fuel source [1,2]. Almost 3.3 Gt of residue (fresh weight) is produced per year in the countries/regions with large biomass potential (EU27, Pan Europe minus the EU27, USA, Canada, Brazil, Argentina, China, and India) from the processing of crops such as wheat, maize, rice, soybean, barley, rapeseed, sugarcane, and sugar beet. Forestry also generates woody biomass residues from timber logging: ~80% of forest tree mass is lost as waste [1,2].

Besides agricultural and forestry waste, the food and drink industry is the major waste contributor. The manufacturing sector of the food and drink industry is the largest in Europe in terms of turnover, added value, and employment. The waste streams from the food and beverages industry in the European Union (EU) have been estimated to be 37 million tons per year [3], whereas the largest contribution is from brewer’s spent grain, sugar beet pulp, grape marc, and a non-negligible amount of 6.98% from vegetable wastes of the fresh-cut sector [4]. Other sources are the potato industry (~50% of the harvested potatoes is waste consisting of peel and other waste) [5], the oranges transformation industry (yielding 3.5 billion pounds of waste, peel, and pulp, per year only in the USA), and tomato waste (~4.4 million tons of tomato waste is produced every year in EU). The output of crustacean aquatic products exceeds 200 million tons per year, which results in ~6–8 million tons of crustacean shells (waste) after processing [6]. Therefore, the conversion of biomass wastes into value-added products has stimulated crescent attention.

Literature reviews have highlighted the importance of recycling biomass wastes, including wood biomass [7,8] and agricultural waste peels [9]. Recent studies have focused on the recycling of natural polymers, such as cellulose, lignin, collagen, gelatin, keratin, and chitin/chitosan, which are extracted from various biomass wastes, including rice husk, peanut shell, corncob, skin, bones of fish, shrimp, and crabs, and then used to fabricate, for example, biomaterials [10], films for packaging [11], and bioactive molecule delivery [12]. However, national and EU regulations for these processes keep control in these fields, allowing the use of only a few solvents, i.e., water (with mixtures of acid or base) and some highly volatile solvents, such as ethyl acetate, ethanol, carbon dioxide (CO_2_), and acetone, whereas the residues are strictly regulated [13,14]. Noteworthy, the value of the products in the industry is not only reliant on the production costs but also on the ways of production; the safety of the process, which ideally would avoid the use of chemicals dangerous to human health and environmentally friendly disposal of the waste products, is a fundamental aspect that should also be considered in calculating the real costs of production. Green technologies are assuming a crescent centrality in modern industrial processing as routes to improve the processes of production while minimizing the impact on the environment. In this field, the selection of solvents plays a critical role in areas such as food, pharmaceutical, and agrochemical processes.

In 2003, the Rogers group reported, for the first time, the processing of unmodified biomass (wood) using ionic liquids [15] (ILs, salts with melting points below 100 °C, often below room temperature [16]). Throughout the following years, multiple ILs were developed as solvents for biopolymers and biomasses. Imidazolium-based ILs possessing basic anions have become the state-of-the-art ILs for this purpose. The most common of these are 1-ethyl-3-methylimidazolim and 1-butyl-3-methylimidazolim acetate and chloride ([C_2_mim][OAc], [C_4_mim][OAc], [C_2_mim]Cl, and [C_4_mim]Cl, respectively) [17,18]. The ability to modify ions, append ion substituents of certain carbon chain length, add co-solvents (e.g., DMSO), etc., permitted achievement of the desired physical and chemical attributes of the systems.

At that time, ILs were defined as compounds composed solely of ions. In protic ILs, this meant a proton transfer from an acid to a base [19], assuming the ΔpK_a_ value (ΔpK_a_ = pK_a_(base) − pK_a_(acid)) was sufficient to shift the equilibrium in the direction of IL formation (ΔpK_a_ > 4) or even produce ILs with near to ideal ionicity (ΔpK_a_ > 8) [20]. Later insights, however, demonstrated that there is a continuum of proton transfer vs. hydrogen bonding at the molecular level. If complete proton transfer took place, this would result in the formation of the IL. However, oftentimes hydrogen bonding between two actives occurs instead with the formation of “liquid co-crystals” [21] or deep eutectic solvent systems (DES) [22,23], sensitive to pH and ionic strength changes.

The DESs, eutectic mixtures of Lewis (or Brønsted) acids and bases with incomplete proton transfer [24], were intentionally designed by choosing two or more distinct components that could interact via hydrogen bonding. The DESs, often with non-stoichiometric ratios between components, present melting points significantly lower than the starting materials [25] and produce mixtures of charged and neutral species. DESs have been proposed for the separation of a wide range of biomolecules and were found to be suitable for biomass waste valorization [26,27,28], including post-harvest treatments in agrochemical uses [29,30] and separation of various biopolymers such as cellulose [31], lignin [32], starch [3,33], and chitin [34]. Comprehensive reviews have covered the increasing literature reporting advances for polymers and chemicals dissolution and extraction from biomass using DES [35]. In our mini-review, we would like to highlight the progress related to chitin extraction, the recovery of chemicals from agri-food waste, and biofuel production using DESs (Figure 1) as examples of biomass-waste valorization.

## 2. Chitin Isolation from Biomass-Waste with DESs

Crustacean waste biomass is the most abundant source of chitin, with a chitin amount of 15–35% [36]. In addition to chitin, crustacean biomass contains 20–40% of proteins and 20–50% of minerals [37,38]. DESs that are liquids at room temperature have been proposed for chitin isolation through separation or dissolution from crustacean biomass. Please note that the words “separation” and “dissolution” are here used distinctly, since DESs can reactively dissolve everything but chitin (a process known as “pulping”) or can dissolve chitin. Typically, choline chloride ([Cho]Cl) or betaine (either betaine, Bet, or betaine hydrochloride, [Bet]Cl) are mixed with urea, thiourea, or weak organic acids (Table 1). The ratio between components plays a vital role as follows: the systems composed of the same components but with different component ratios have been reported to either dissolve or pulp chitin polymer. In the following paragraphs, we will discuss the application of DES for chitin recovery from biomass through both pulping and dissolution. For chitin recovery, traditional heating (usually oil-bath) is normally used, contrary to the microwave irradiation typically employed in the case of chitin extraction using ILs. We would like to point out that other processes such as the dissolution of pure chitin and its surface modification and/or use to produce nanomaterials were recently extensively and comprehensively reviewed [39,40,41,42].

*Chitin Pulping.* In chitin pulping from biomass, protein removal is accomplished by treatment of chitin with an alkaline component of DESs at an elevated temperature, whereas mineral removal is achieved by reaction of minerals with the DES acidic component, forming salts, water, and CO_2_ gas. Following this approach, the system [Cho]Cl/Lactic acid (1:2.5 mol/mol ratio) was reported to be the best one (highest deproteinization efficiency) and suitable for recovery of high purity chitin (99.33%) from shrimp shells (Table 1, entry #1) [43]. The crystallinity (CrI) of the recovered chitin was reported to be slightly lower than for the commercial chitin (80.32 vs. 85.49%, respectively), which was attributed to the acylation of chitin (Degree of Substitution (DS) of 0.43) [43]. In fact, the acylation of chitin after pulping with DES is commonly reported, especially when organic acids are used in molar excess as part of the DES composition. Thus, *O*-acylated chitin was isolated directly from shrimp shells by mixing shrimp shells (0.25 g) and DES (5 g) and heating the mixture at 150 °C for 3 h (Table 1, entry #2) [44]. The heterogeneous acylation of chitin was only carried out when the mole ratio of the [Cho]Cl/organic acid was 1:2. With [Cho]Cl/Malic acid (1:2 mol/mol), the purity of *O*-malate chitin (13.2% yield) reached 98.6% with a DS of 0.46, and the product exhibited antibacterial and anti-tumor effects.

Chitin was also recovered from lobster shells using [Cho]Cl/Malonic acid (1:2 mol/mol ratio, 2 h at 50 °C), achieving 16.2% chitin yield with 80.6% CrI (Table 1, entry #3) [45]. The CrI of the resulting chitin decreased depending on the DES system used as follows: [Cho]Cl/Urea (43%) < [Cho]Cl/Citric acid (76%) < [Cho]Cl/Malonic acid (81%) < [Cho]Cl/Lactic acid (98%). The same trend was observed for molecular weight (Mw) as follows: [Cho]Cl/Urea (75 kDa) < [Cho]Cl/Citric acid (84 kDa) < [Cho]Cl/Malonic acid (96 kDa) < [Cho]Cl/Lactic acid (125 kDa). Such low Mw is likely due to the fact that, in the case of pulping, chitin is degraded due to glycosidic bond hydrolysis [17,46]. When compared with the [C_2_mim][OAc]-extraction process, the latter is a physical, dissolution process as follows [47,48]: the inter- and intramolecular hydrogen bonds get disrupted [49], and the IL dissolves the polymer in its “native” state, resulting in a significantly larger Mw [49,50,51,52]. On the other hand, DESs produce a polymer of notably higher crystallinity than the IL-extraction, for the same reason. Another important difference is that pulping results in chitin deacetylation (%DDA) to a significant extent (7–11%), whereas with the IL extraction %DDA is nearly ideal (2 ± 1%) [53].

Another example includes chitin pulping with [Cho]Cl/Maleic acid (1:1 mol/mol) DES using microwave irradiation (Table 1, entry #4) [54]. Crustacean biomass was mixed with DES at 5–20 wt% load and heated under 700 W microwave irradiation for different periods of time (1–9 min), with 3 s pulses. Then, chitin was separated from the DES by centrifugation. The obtained chitin was free from proteins and minerals, and highly crystalline (CrI 71%), although the most important characteristics of chitin—Mw and %DA—were not determined [54]. This same research group proposed the recovery of chitin from crab shells using [Cho]Cl/Malic acid (1:1 mol/mol) at crab shell/DES ratios of 1:10, 1:20, and 1:30 and heated under 700 W microwave irradiation for different periods of time (3–11 min) (Table 1, entry #5) [55]. The maximal demineralization and deproteinization efficiency reached 99.8% and 92.3%, respectively, at the crab shell/DES ratio of 1:30 after 11 min of microwave irradiation [55]. While most of these examples have not evaluated the state of chitin, several sources claim that pulped chitin exists in the nanocrystalline form [56].

Recently, [Cho]Cl/Glycerine (1:2 mol/mol) was proposed to separate chitin from shrimp shell powder using ball-milling. After ball-milling the DES and shrimp shell for 3 h at 100 °C, acetic acid was added, resulting in a 25.8% yield after discoloration (Table 1, entries #6–12) [57]. When the acetic acid concentration was raised from 2.5 to 10.0 wt%, the chitin yields decreased gradually from 28.5 to 21.0%. These yield changes were ascribed to the acidic hydrolysis of residual protein and the decomposition of calcium carbonate. Although the addition of acetic acid did not affect the %DA, which remained stable above 80%, the Mw of the chitin decreased from 270 to 190 kDa with the increase of acid from 2.5 to 10.0 wt%.

The two-step approach was also proposed by Zhao et al., who used citric acid to remove minerals from shrimp shells, achieving 98% demineralization efficiency, followed by the removal of protein using deep eutectic solvents with the assistance of microwave, reaching 88% deproteinization efficiency (Table 1, entries #13–16) [58]. The maximal deproteinization rates of [Bet]Cl/urea (1:2 mol/mol), [Cho]Cl/Urea (1:2 mol/mol), [Cho]Cl/Ethylene glycol (1:2 mol/mol), and [Cho]Cl/Glycerol (1:2 mol/mol) reached 93, 92, 90.6, and 88.6%, respectively. The yields of chitin extracted by [Bet]Cl/urea, [Cho]Cl/Urea, [Cho]Cl/Ethylene glycol, and [Cho]Cl/Glycerol were 23.6, 25.1, 24.8, and 22.5%, respectively, which were higher than that of the acid/alkali-extracted chitin (17.7%).

Although crustaceous shells are the main target for chitin sources, their recovery from alternative sources using DES has recently been reported. Skimmed black soldier fly prepupae powder (5 g) was mixed with 50 g DESs and heated in a rotary shaker at different temperatures of 50 °C and 80 °C for 2 h (Table 1, entries #17–36) [59]. The best chitin yields were obtained with Bet/Urea (1:1 mol/mol) and [Cho]Cl/Urea (1:2 mol/mol), i.e., 26.7 and 26.0%, respectively. The proton (H^+^) released from hydrogen bond acceptor (HBA) or hydrogen bond donor (HBD) was the main reason for demineralization, whereas intermolecular and intramolecular hydrogen bonds in DESs facilitated the removal of proteins [43,59].

DESs were also used to pre-treat white button mushrooms to prepare chitin-glucan complexes (CGC), achieving yields in the range of 23.5–30.4% of raw material (mushroom powder, data not shown in Table 1) [60]. Chitin in mushrooms is conjoined to *β*-glucan; such a CGC may have advantages arising from both chitin and *β*-glucan in terms of health benefits and structural properties [60]. The CGCs yields were significantly higher than using the traditional basic process, 17.0 wt%), with high levels of chitin (17.1–23.2 wt%) and *β*-glucan (23.2–32.8 wt%).

*Chitin Dissolution.* All systems that were shown to be efficacious for chitin dissolution had urea in their composition, and the amount of urea was quite significant. Chitin dissolution has been explained by the breakage of hydrogen bonds between the biopolymer chains and their replacement with new hydrogen bonds between chitin and [Cho]Cl, as well as by the reaction between the acetamido group in chitin and the free H^+^ ions of the hydroxyl group of choline cation in [Cho]Cl [34]. Chitin was isolated by mixing 1 or 2 g shrimp shells powder with 50 g of DES ([Cho]Cl combined with Lactic, Malonic, or Citric acids) at elevated temperature (60–90 °C) to create a suspension (Table 1, entries #37–40) [34]. After the shrimp biomass dissolution, water was added to the suspension and the precipitated chitin was filtered. The biomass load is usually relatively low, from 2 to 4 wt%. The highest chitin yield, 85% of that available, was obtained using [Cho]Cl/Lactic acid (1:1 mol/mol) DES. The purity of the resulting chitin varied from 95 to 98%; again, the highest purity was obtained using [Cho]Cl/Lactic acid (1:1 mol/mol) while the lowest purity, 95%, was obtained using [Cho]Cl/Urea (1:2 mol/mol).

A large number of eutectics, made from [Cho]Cl or Bet and weak organic acids, were screened for the isolation of chitin from crayfish shells [61]. The systems were screened for their ability to dissolve chitin included: Bet/Formic acid (1:2 mol/mol), Bet/Lactic acid (1:2 mol/mol), Bet/Acetic acid (1:2 mol/mol), Bet/Propionic acid (1:2 mol/mol), Urea/Lactic acid (1:2 mol/mol), Urea/Propionic acid (1:2 mol/mol), Bet/Lactic acid (1:2 mol/mol), Urea/Formic acid (1:1.5 mol/mol), Urea/Acetic acid (1:1.5 mol/mol), [Cho]Cl/Formic acid (1:2 mol/mol), [Cho]Cl/Lactic acid (1:2 mol/mol), [Cho]Cl/Propionic acid (1:2 mol/mol), and [Cho]Cl/Acetic acid (1:2 mol/mol). Crayfish shells were mixed with DESs (5 wt%), stirred, and heated in an oil bath at 115 °C for 20 h. After that, water was added to precipitate chitin dissolved in eutectics, and chitin was filtered out of the solution. [Cho]Cl/Lactic acid (1:2 mol/mol), [Cho]Cl/Formic acid (1:2 mol/mol), and Bet/Lactic acid (1:2 mol/mol) dissolved the largest amount of chitin (9–11%). Based on the results, [Cho]Cl/Lactic acid (1:2 mol/mol) and Bet/Lactic acid (1:2 mol/mol) were chosen as the most appropriate solvents for the extraction of chitin from crayfish shell wastes (Table 1, entries #41–42). The chitin yield was found to be 85 ± 1% for both DESs, based on chitin available in biomass [61].

Similarly, Gluconic acid/Amino acids DESs (aqueous solutions) [62] allowed the isolation of chitin from the shell of a snow crab (*Chionoecetes opilio*) with the best performance reported from Gluconic acid/Cysteine DES. The Mw, yield, and purity of chitin were 3.75 × 10^5^ Da, 79.1 and 94.5%, respectively, under the optimal conditions (100 °C for 6 h with a solid-to-liquid ratio of 1:20) (Table 1, entries #43–61). The yield of chitin obtained by Gluconic acid/Cysteine was 1.35 times higher than that obtained by the traditional acid-alkali pulping [62]. This same group also carried out chitin isolation from the snow crab biomass by its extraction into the DESs (1:20 shell/DESs ratio) and heating at 130 °C for 3 h [63]. After the reaction, water was added to the mixture and centrifuged to obtain a chitin product (Table 1, entries #62–79). The highest purity (93.4%) of chitin came from [Cho]Cl/Formic acid, while the highest yields were achieved with the systems [Cho]Cl/*N*-acetyl-D-glucosamine (90.6%) and Bet/*N*-acetyl-D-glucosamine (90.7%), although with low purity (~60%). The purity of chitin prepared from Bet-based DESs was slightly lower than that of [Cho]Cl-based DESs, probably due to the higher viscosity of Bet-based DESs. To achieve high yields with high purity, ternary systems based on [Cho]Cl, *N*-acetyl-D-glucosamine, and formic acid were prepared using different ratios. Using [Cho]Cl/*N*-acetyl-D-glucosamine/Formic acid (1:0.6:1.4 mol/mol/mol), a chitin with medium Mw of 3.92 × 10^5^ Da, 90.2% purity, and 85.6% yield was obtained [63].

**Table 1 molecules-27-06606-t001:** Deep eutectic systems reported for chitin recovery from biomass.

#	DES System	Composition, mol/mol	Conditions	Time, h	Yield, %	References
**Isolation of Chitin by Pulping**						
1	[Cho]Cl/Lactic acid	1:2.5	Stirring/Heating at 150 °C	6	- ^a^	[43]
2	[Cho]Cl/Malic acid	1:2	Stirring/Heating at 150 °C	3	13.2 ^b^	[44]
3	[Cho]Cl/Malonic acid	1:2	Stirring/Heating at 50 °C	2	16.2 ^b^	[45]
4	[Cho]Cl/Maleic acid	1:1	Microwaving (700 W)	0.15	- ^a^	[54]
5	[Cho]Cl/Malic acid	1:1	Microwaving (700 W)	0.18	- ^a^	[55]
6	[Cho]Cl/Glycerol + 7.5% Acetic acid	1:2	Stirring/ heating at 100 °C	3	25.8 ^b^	[57]
7	[Cho]Cl/Glycerol + 7.5% Acetic acid	1:2	Stirring/ heating at 80 °C	3	25.8 ^b^	[57]
8	[Cho]Cl/Glycerol + 7.5% Acetic acid	1:2	Stirring/ heating at 120 °C	3	21.4 ^b^	[57]
9	[Cho]Cl/Glycerol + 7.5% Acetic acid	1:2	Stirring/ heating at 140 °C	3	21.5 ^b^	[57]
10	[Cho]Cl/Glycerol + 2.5% Acetic acid	1:2	Stirring/ heating at 120 °C	3	28.5 ^b^	[57]
11	[Cho]Cl/Glycerol + 5.0% Acetic acid	1:2	Stirring/ heating at 120 °C	3	24.3 ^b^	[57]
12	[Cho]Cl/Glycerol + 10% Acetic acid	1:2	Stirring/ heating at 120 °C	3	21.0 ^b^	[57]
13	[Bet]Cl/Urea	1:2	(1) 10% citric acid for demineralization (2) DES + microwave irradiation	0.15	23.6 ^b^	[58]
14	[Cho]Cl/Urea	1:2	(1) 10% citric acid for demineralization (2) DES + microwave irradiation	0.15	25.1 ^b^	[58]
15	[Cho]Cl/Ethylene glycol	1:2	(1) 10% citric acid for demineralization (2) DES + microwave irradiation	0.15	24.8 ^b^	[58]
16	[Cho]Cl/Glycerol	1:2	(1) 10% citric acid for demineralization (2) DES + microwave irradiation	0.15	22.5 ^b^	[58]
17	[Cho]Cl/DL-Lactic acid	1:2	Stirring/Heating at 50 °C	2	23.3 ^b^	[59]
18			Stirring/Heating at 80 °C	2	16.4 ^b^	[59]
19	[Cho]Cl/Urea	1:2	Stirring/Heating at 50 °C	2	26.0 ^b^	[59]
20			Stirring/Heating at 80 °C	2	22.8 ^b^	[59]
21	[Cho]Cl/Glycerol	1:2	Stirring/Heating at 50 °C	2	16.7 ^b^	[59]
22			Stirring/Heating at 80 °C	2	22.9 ^b^	[59]
23	[Cho]Cl/*n*-butyric acid	1:2	Stirring/Heating at 50 °C	2	22.3 ^b^	[59]
24			Stirring/Heating at 80 °C	2	12.0 ^b^	[59]
25	[Cho]Cl/Oxalic acid	1:2	Stirring/Heating at 50 °C	2	23.3 ^b^	[59]
26			Stirring/Heating at 80 °C	2	12.7 ^b^	[59]
27	Bet/Urea	1:1	Stirring/Heating at 50 °C	2	26.7 ^b^	[59]
28			Stirring/Heating at 80 °C	2	12.0 ^b^	[59]
29	Bet/Glycerol	1:2	Stirring/Heating at 50 °C	2	25.5 ^b^	[59]
30			Stirring/Heating at 80 °C	2	22.8 ^b^	[59]
31	Bet/*n*-butyric acid	1:2	Stirring/Heating at 50 °C	2	24.5 ^b^	[59]
32			Stirring/Heating at 80 °C	2	14.5 ^b^	[59]
33	Bet/Oxalic acid	1:2	Stirring/Heating at 50 °C	2	22.9 ^b^	[59]
34			Stirring/Heating at 80 °C	2	6.5 ^b^	[59]
35	Bet/DL-Lactic acid	1:2	Stirring/Heating at 50 °C	2	25.7 ^b^	[59]
36			Stirring/Heating at 80 °C	2	14.3 ^b^	[59]
**Isolation of Chitin by Dissolution**						
37	[Cho]Cl/Urea	1:1	Heating at 60–90 °C	1.67	– ^a^	[34]
38	[Cho]Cl/Citric acid	1:1	Heating at 60–90 °C	1.67	– ^a^	[34]
39	[Cho]Cl/Malonic acid	1:1	Heating at 60–90 °C	1.67	– ^a^	[34]
40	[Cho]Cl/Lactic acid	1:1	Heating at 70 °C	1.67	85 ^c^	[34]
41	Bet/Lactic acid	1:2	Stirring/ heating in an oil bath at 115 °C	20	85 ^c^	[61]
42	[Cho]Cl/Lactic acid	1:2	Stirring/ heating in an oil bath at 115 °C	20	85 ^c^	[61]
43	Gluconic Acid/Aspartic acid	5:1	Stirring/ heating in an oil bath at 90 °C	24	71.2 ^c^	[62]
44	Gluconic Acid/Threonine	5:1	Stirring/ heating in an oil bath at 90 °C	24	74.3 ^c^	[62]
45	Gluconic Acid/Serine	5:1	Stirring/ heating in an oil bath at 90 °C	24	76.3 ^c^	[62]
46	Gluconic Acid/Glutamic acid	5:1	Stirring/ heating in an oil bath at 90 °C	24	71.7 ^c^	[62]
47	Gluconic Acid/Proline	5:1	Stirring/ heating in an oil bath at 90 °C	24	74.9 ^c^	[62]
48	Gluconic Acid/Glycine	5:1	Stirring/ heating in an oil bath at 90 °C	24	74.7 ^c^	[62]
49	Gluconic Acid/Alanine	5:1	Stirring/ heating in an oil bath at 90 °C	24	72.4 ^c^	[62]
50	Gluconic Acid/Cysteine	5:1	Stirring/ heating in an oil bath at 90 °C	24	71.3 ^c^	[62]
51	Gluconic Acid/Valine	5:1	Stirring/ heating in an oil bath at 90 °C	24	72.9 ^c^	[62]
52	Gluconic Acid/Methionine	5:1	Stirring/ heating in an oil bath at 90 °C	24	71.1 ^c^	[62]
53	Gluconic Acid/Isoleucine	5:1	Stirring/ heating in an oil bath at 90 °C	24	71.9 ^c^	[62]
54	Gluconic Acid/Leucine	5:1	Stirring/ heating in an oil bath at 90 °C	24	70.9 ^c^	[62]
55	Gluconic Acid/Phenylalanine	5:1	Stirring/ heating in an oil bath at 90 °C	24	72.7 ^c^	[62]
56	Gluconic Acid/Lysine	5:1	Stirring/ heating in an oil bath at 90 °C	24	80.4 ^c^	[62]
57	Gluconic Acid/Histidine	5:1	Stirring/ heating in an oil bath at 90 °C	24	79.7 ^c^	[62]
58	Gluconic Acid/Arginine	5:1	Stirring/ heating in an oil bath at 90 °C	24	80.0 ^c^	[62]
59	Gluconic Acid/Tryptophan	5:1	Stirring/ heating in an oil bath at 90 °C	24	72.6 ^c^	[62]
60	Gluconic Acid/Asparagine	5:1	Stirring/ heating in an oil bath at 90 °C	24	78.5 ^c^	[62]
61	Gluconic Acid/Glutamine	5:1	Stirring/ heating in an oil bath at 90 °C	24	80.1 ^c^	[62]
62	[Cho]Cl/*N*-acetyl-d-Glucosamine	2:1	Stirring/ heating at 130 °C	3	90.6 ^c^	[63]
63	[Cho]Cl/D-Gluconic acid	1:2	Stirring/ heating at 130 °C	3	82.7 ^c^	[63]
64	[Cho]Cl/Hydroxymethylfurfural	1:2	Stirring/ heating at 130 °C	3	85.6 ^c^	[63]
65	[Cho]Cl/Levulinic acid	1:2	Stirring/ heating at 130 °C	3	75.0 ^c^	[63]
66	[Cho]Cl/Acetic acid	1:2	Stirring/ heating at 130 °C	3	71.5 ^c^	[63]
67	[Cho]Cl/Formic acid	1:2	Stirring/ heating at 130 °C	3	66.2 ^c^	[63]
68	Bet/*N*-acetyl-d-Glucosamine	2:1	Stirring/ heating at 130 °C	3	90.7 ^c^	[63]
69	Bet/D-Gluconic acid	1:2	Stirring/ heating at 130 °C	3	84.2 ^c^	[63]
70	Bet/5-Hydroxymethylfurfural	1:2	Stirring/ heating at 130 °C	3	86.2 ^c^	[63]
71	Bet/Levulinic acid	1:2	Stirring/ heating at 130 °C	3	75.4 ^c^	[63]
72	Bet/Acetic acid	1:2	Stirring/ heating at 130 °C	3	71.2 ^c^	[63]
73	Bet/Formic acid	1:2	Stirring/ heating at 130 °C	3	70.1 ^c^	[63]
74	[Cho]Cl/*N*-acetyl-d-Glucosamine/Formic acid	1:0.2:1.8	Stirring/ heating at 130 °C	3	74.3 ^c^	[63]
75	[Cho]Cl/*N*-acetyl-d-Glucosamine/Formic acid	1:0.6:1.4	Stirring/ heating at 130 °C	3	85.6 ^c^	[63]
76	[Cho]Cl/*N*-acetyl-d-Glucosamine/Formic acid	1:1:1	Stirring/ heating at 130 °C	3	88.2 ^c^	[63]
77	[Cho]Cl/*N*-acetyl-d-Glucosamine/Formic acid	1:1.4:0.6	Stirring/ heating at 130 °C	3	89.6 ^c^	[63]
78	[Cho]Cl/*N*-acetyl-d-Glucosamine/Formic acid	2:0.9:0.1	Stirring/ heating at 130 °C	3	92.2 ^c^	[63]
79	[Cho]Cl/*N*-acetyl-d-Glucosamine/Formic acid	2:0.7:0.3	Stirring/ heating at 130 °C	3	90.7 ^c^	[63]

^a^ Yield value not reported. ^b^ Percent yield is reported based on weight of starting material. ^c^ Percent yield is reported based on chitin available in biomass.

## 3. Deep Eutectic Solvents for the Recovery of Chemicals from Agri-Food Industrial Waste

In the extraction of phenolic compounds, DESs are employed mainly in the separation of carotenoids, flavonoids, and pigments with a wide range of polarity from lignocellulose. In the area of recovery of agri-food industrial by-products, a variety of [Cho]Cl-based DESs were used to extract phenolic compounds from virgin olive oils. Two of the DESs, [Cho]Cl/Xylitol and [Cho]Cl/1,2-Propanediol, showed an increase of extraction yield up to 20–33% and 67.9–68.3% compared to a conventional system, 80 vol% methanol/water [64]. In 2017, a study about the ultrasound-assisted [Cho]Cl/Malic acid extraction of wine lees anthocyanins was reported [65]. In this study, the optimum time conditions were 30.6 min extraction time, 341.5 W ultrasound power, and 35.4 wt% water content in the DES. Grudniewska et al., used [Cho]Cl/Glycerol for enhanced extraction of proteins from oilseed cakes [66]. They extracted the proteins into the DES, then the extract was precipitated upon the addition of water.

Fernandez et al., employed glucose/lactic acid, glucose/citric acid, and fructose/citric acid in 5:1, 1:1, and 1:1 mol/mol ratios, respectively, for ultrasound-based extraction of 14 phenolic compounds from onion, olive, tomato, and pear by-products at 40 °C [67]. The aqueous glucose/lactic DES resulted as the optimal solvent with the highest capacity of extraction, comparable to those of methanol and water. It was concluded that the glucose/lactic acid DES yielded higher extractability.

Deng et al., synthesized a series of water-soluble DESs composed of Hexafluoroisopropanol (HFIP) as HBD and L-Carnitine or Bet as HBAs to extract pyrethroid residues from tea beverages and fruit juices [68]. The extraction method based on L-Carnitine/HFIP (1:2 mol/mol) solvent showed several advantages, such as a short extraction time and a high enrichment factor. Eutectic mixtures can perform as reaction media or extraction solvents for the bioconversion of several components. The applicability of DESs for the removal of cadmium from rice flour was examined by Huang et al. [29]. Among the [Cho]Cl-based and glycerol-based DESs, the former demonstrated good removal of Cd (51–96%). The interesting point was that the DESs did not affect the structure or chemical components of rice flour.

Di Gioia et al., explored the possibility of a selective conversion of furfural, produced by biomass, to biofunctionalized cyclopentenone derivatives in [Cho]Cl/Urea [69]. In another study, cellulose derived from sunflower stalks was converted into value-added components [70]. Three DESs, namely, [Cho]Cl/Oxalic acid, [Cho]Cl/Citric acid, and [Cho]Cl/Tartaric acid, were used as solvents and catalysts. With [Cho]Cl/Oxalic acid-based DES under microwave irradiation, ~99% carbon efficiency was obtained at 180 °C in 1 min. Under such conditions, 4.1% of 5-hydroxymethyl furfural (5-HMF), 76.2% of levulinic acid, 5.6% of furfural, and 15.2% of formic acid were isolated.

## 4. Deep Eutectic Solvents for Biofuels Production

The awareness of the significance of biomass contribution to energy consumption is increasing as follows: For instance, the EU has indicated that ~10% of the entire gross final-energy consumption in the EU must come from biomass, while the United States of America has set itself an ambitious goal aiming to achieve a total biofuel output of 136.3 billion liters by 2022 [1]. However, introducing new processing strategies within the biorefinery context can be challenging because of the typical discrepancy between the conditions used for pretreatment and those used downstream for saccharification and fermentation. For example, pretreatments under acidic or basic conditions are usually not compatible with downstream processing (e.g., enzymatic saccharification and microbial fermentation) and require neutralization and/or separation stages before proceeding with the next steps.

Delignification is the initial pretreatment step for biofuel production. It is a chemically intensive and environmentally problematic process [71]. Two of the most widely used methods are Kraft and OrganoSolv delignification. The Kraft process employs a hot mixture of water, sodium hydroxide, and sodium sulfide [72], and requires lots of energy to reduce sulfide-containing black liquor waste emissions. Handling this waste poses hazards to the environment [73]. The second type of pulping, OrganoSolv, uses catalysts in organic solvents, and the OrganoSolv family includes ASAM (alkali-sulfite-anthraquinone-methanol) [74], Organocell (sodium hydroxide-methanol-anthraquinone) [75], Formacell (acetic acid-formic acid) [76,77], Milox (multistage peroxyacid treatment) [78], etc. High capital and processing costs are associated with solvent recycling [79], and none of these processes has been permanently applied on an industrial scale [80]. Since the discovery of ILs’ biomass processing ability, the delignification of biomass using ILs has been reported [81,82,83,84,85], and there are multiple ILs that have been proposed as good solvents for lignin but not cellulose. Variables such as type of biomass, its load, time and temperature, particle size, etc. affect delignification and delignification rates [86]. The ILs are often used with a co-solvent and/or a catalyst (e.g., polyoxymetallates POMs [86]). Recently, a [C_4_mim][HSO_4_]/butanediol/water system was demonstrated to delignify wood to as low as <1% lignin content with an efficiency of 98% [87].

The DESs have also become widely acknowledged as eco-friendly delignification systems. Efficient delignification was reported using [Cho]Cl/lactic acid [88,89,90], propionic acid/Urea [89], [Cho]Cl/p-Toluenesulfonic acid [89], [Cho]Cl/formic acid [91], [Cho]Cl/acetic acid [91], [Cho]Cl/oxalic acid [92], [Cho]Cl/malic acid [91], etc. [93,94,95,96]. Due to their ability to remove lignin and xylan, DES such as [Cho]Cl/formic acid or triethylbenzylammonium chloride/lactic acid were proposed for the pretreatment of corn stover to produce biobutanol via acetone-butanol-ethanol (ABE) fermentation by *Clostridium* [97,98,99]. A similar approach, i.e., using [Cho]Cl-based DES for biomass, Bambara groundnut haulm pretreatment, was recently proposed for bioethanol production. Thus, [Cho]Cl/Lactic acid pretreatment at 100 °C for 1 h was observed to be the best condition for hemicellulose (54.5%) and lignin (60.7%) removal, along with optimum sugar recovery of 94.8% [100]. The resulting hydrolysate was concentrated, washed, and fermented for 72 h with *Saccharomyces cerevisiae* BY4743, and a maximum ethanol concentration of 11.57 g/L was achieved with an ethanol yield of 0.38 g/g sugar and productivity of 0.19 g/L/h. However, in these works, the rationale for the selection of the DES for this specific application was not discussed. Considering that their application was proposed in batches, with separation, filtration, and neutralization steps between the pre-treatment, saccharification, and fermentation steps, and that the number of available DESs is high [35], a comprehensive DES screening might be of value to identify the best type of DESs for in-batch biofuel production.

The recent development of biocompatible ILs, such as choline lysinate or ethanolamine acetate, allows for a consolidated one-pot biomass-to-biofuel conversion process that combines pretreatment, saccharification, and fermentation in one vessel [101,102,103]. Similarly, biocompatible DESs were evaluated for the conversion of biomass into biofuels and bioproducts using a one-pot process, an approach that can reduce the operating costs because it simplifies process design and reduces the energy input (avoids the mass transfer between reactors) (Figure 2) [104]. A multistep ethanol conversion from corn stover was demonstrated in a single vessel as follows: First, the biomass was treated with [Cho]Cl/Glycerol (1:2 mol/mol) at 50 °C for 24 h, followed by a simultaneous saccharification and yeast fermentation at 37 °C for 48 h. Compared to conventional configurations, the one-pot process eliminated all solid/liquid separation steps and did not require any pH adjustment. The process generated 134 g of ethanol from 1 kg of corn stover, which is equal to a conversion yield of 77.5% based on the glucose present [104].

The anaerobic digestion of biological and food wastes produces biogas, which is considered a renewable energy supply. Biogas’ main impurity is CO_2_, which should be removed in the upgrading process. In a very recent study, the DES [Cho]Cl/Urea, in an aqueous form, was employed as a liquid absorbent in a conceptual process to upgrade biogas [105]. In comparison with a pure water process, the DES addition decreased energy use by 16%. Moreover, to study how the environment could be influenced by the process, they employed the Green Degree (GD) assessing method [106]. The DES loss was negligible due to its very low vapor pressure and thermal stability. They found that the calculated difference of GDs was higher than zero for aqueous [Cho]Cl/Urea-based processes, demonstrating that this process is environmentally benign.

## 5. Conclusions and Perspectives

In the past decade, significant advances based on DESs have been demonstrated to add value to biomass-waste streams as alternatives to ILs. In fact, the name DES was invented because the reviewer of the very first paper advised against calling them ILs [107]. Nonetheless, the development of DESs has an enormous development potential, and there is no doubt that the DESs are rich in terms of application prospects. The straightforward design of natural-based DESs is a major advantage for their application and potential implementation in transformative processes.

DESs can be applied in diverse ways, and the structures and applications of DESs will become increasingly more versatile, and new manufacturing technologies based on DESs will soon be developed. There are a few questions that remain unanswered, and these questions are the same as those for ILs. Thus, the cytotoxicity and/or toxicity of DESs were shown to be higher than those of their constituents [108]. It has also been reported that high temperature, microwave- or ultrasound- irradiation potentially leads to DESs decomposition, which might present a challenge during the recovery [109]. The number of potential reuses also remains unknown. We must also point out the challenge of scaling-up DES-based processes, mostly due to the necessity of DES recycling. Although advertised as recyclable, DESs will be contaminated with by-products in the case of complex biomass (e.g., proteins and minerals in the case of crustacean biomass).

Importantly, the same overgeneralizations as those for ILs hunt the field of DESs as follows: People report that DES systems are green, non-volatile, and non-toxic, but again, DESs are a class of compounds and can have any of the properties as we designed them to have. As chemists, we should stop looking for advantages or disadvantages of ILs vs DESs, presenting them as “competitors”. The reality is that both ILs and DESs are available when choosing solvents for a specific process. Understanding their differences from a chemical perspective will allow us to select the best one for each application based on performance, availability, cost, environmental impact, etc.

The constantly crescent request for more sustainable (bio)chemical processes will address the scientific community to deeper investigate these solvents and develop new systems and process technologies in addition to the classical extraction methods, both to optimize the production processes and allow the effective transformation of biomass into chemicals or polymers in the typical circular approach of resource utilization.

## Figures and Tables

**Figure 1 molecules-27-06606-f001:**
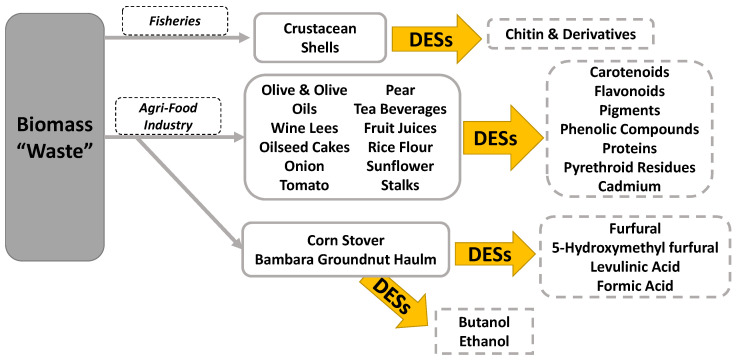
Overview of the examples of biomass-based waste treated with DESs for its valorization discussed in this review.

**Figure 2 molecules-27-06606-f002:**
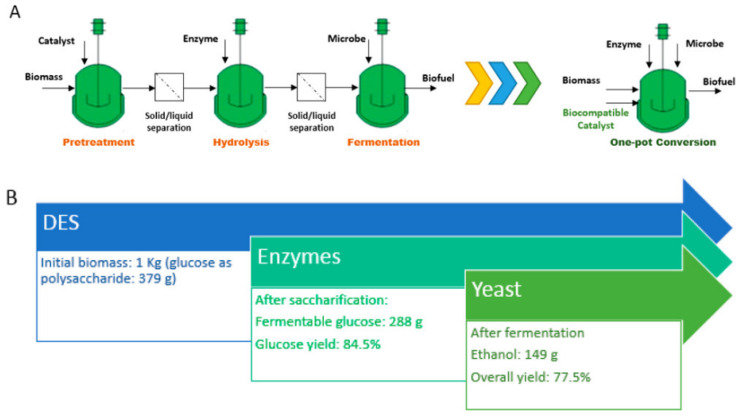
(**A**) One-pot bioethanol production with biocompatible deep eutectic solvents ([Cho]Cl/Glycerol, 1:2 mol/mol) and (**B**) glucose balance of the one-pot bioethanol conversion. Reprinted with permission from Ref. [104]. Copyright 2018, American Chemical Society.

## Data Availability

Not applicable.

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
