# Peer review of "Deep Eutectic Solvents: Alternative Solvents for Biomass-Based Waste Valorization"

_molecules, 2022, doi:10.3390/molecules27196606_

Round 1
Reviewer 1 Report
This review has summarized the application of deep eutectic solvents (DES) in the field of biomass-based treatments, focusing on the three examples such as chitin recovery from biomass, and isolation of valuable chemicals and biofuels from biomass-waste streams. It is of significance to broaden the windows of DES application. The review is relatively comprehensive and organized well. Thus, it can be considered publication after resolving the following issues.
1. Page 2: Ionic liquids was mentioned in the biomass-based treatments, so I suggest that the authors should clearly demonstrate the advantages of DES when compared to ionic liquids.
2. Page 3: The sentence “…Please note, that the word “separation” is used here rather than “dissolution”, because generally there are two types of DESs: those that dissolve chitin, and those that reactively dissolve everything but chitin (process, known as “pulping”).” Please the authors elaborate the difference between “dissolution” and “separation” in the DES application.
3. The en dash "–" instead of "-" should be used between the numbers representing the range. Please check the full text and correct them.
4.Page 3: The CrI of the resulting chitin decreased depending on the DES system used: [Cho]Cl/Urea (43%) < [Cho]Cl/Citric acid (76%) < [Cho]Cl/Malonic acid (81%) < [Cho]Cl/Lactic acid (98%). The same trend was observed for molecular weight (Mw): [Cho]Cl/Urea (75 kDa) < [Cho]Cl/Citric acid (84 kDa) ~ [Cho]Cl/Malonic acid (96 kDa) < [Cho]Cl/Lactic acid (125 kDa). What does the "~"mean in this sentence?
5. Page 3: Another important difference is that pulping results in chitin deacetylation (%DDA) to a significant extent (711%), whereas with the IL extraction %DDA nearly ideal (2(1))%.54 What does (2(1))% mean?
6. The resolution of Figure 1 should be enhanced. In addition, I suggest that the authors should be further increase the number of pictures in main text for the clearer exhibition for readers.
7. From author's point view, what are the main problems to limit the use of DESs in industrial applications such as recovery of beneficial substances from biomass-based waste?
Author Response
- Page 2: Ionic liquids was mentioned in the biomass-based treatments, so I suggest that the authors should clearly demonstrate the advantages of DES when compared to ionic liquids.
Response: We do not see DESs and ILs as competitors but instead as alternative solvents. In fact, often it is difficult to distinguish between the two (see for example, https://doi.org/10.1073/pnas.1815526115). On a molecular level, they differ in nanoscale spatial ordering and extent of ionicity. These systems offer different solutions and understanding their differences from a chemical perspective will allow us to select the best system for each application. The choice would be based on performance, availability, cost, environmental impact, etc. For instance, recent paper (https://doi.org/10.1021/acs.jced.0c00750) screened 14 ILs and 7 DESs for CO2 capture, providing robust characterization of the systems’ performance, and for this application, ILs performed better. On the other side, DESs are more suitable for preparation of biomaterials, where biopolymer dissolution is unjustified (https://www.sciencedirect.com/science/article/abs/pii/S0926669022005295). We have included this point of view in the Conclusions section.
Revision: (Lines 383-387) “As chemists, we should stop looking for advantages or disadvantages of ILs vs DESs, presenting them as “competitors”. The reality is that both ILs and DESs are available when choosing solvents for a specific process. Understanding their differences from a chemical perspective will allow us to select the best one for each application, based on performance, availability, cost, environmental impact, etc.”
- Page 3: The sentence “…Please note, that the word “separation” is used here rather than “dissolution”, because generally there are two types of DESs: those that dissolve chitin, and those that reactively dissolve everything but chitin (process, known as “pulping”).” Please the authors elaborate the difference between “dissolution” and “separation” in the DES application.
Response: The sentence was confusing as was written. The sentence was re-written to indicate the difference of the different processes based on DES.
Revision: (Lines 103-105) “Please note, that the word “separation” and “dissolution” are here used distinctly, since DESs can both reactively dissolve everything but chitin (process, known as “pulping”) or can dissolve chitin.”
- The en dash "–" instead of "-" should be used between the numbers representing the range. Please check the full text and correct them.
Response: The dash was changed within the entire manuscript for consistency.
- Page 3: The CrI of the resulting chitin decreased depending on the DES system used: [Cho]Cl/Urea (43%) < [Cho]Cl/Citric acid (76%) < [Cho]Cl/Malonic acid (81%) < [Cho]Cl/Lactic acid (98%). The same trend was observed for molecular weight (Mw): [Cho]Cl/Urea (75 kDa) < [Cho]Cl/Citric acid (84 kDa) ~ [Cho]Cl/Malonic acid (96 kDa) < [Cho]Cl/Lactic acid (125 kDa). What does the "~"mean in this sentence?
Response: The symbol indicated “approximately equal” but it is now changed to “<” to better reflect the trend.
Revision (Lines 136-140): “The CrI of the resulting chitin decreased depending on the DES system used: [Cho]Cl/Urea (43%) < [Cho]Cl/Citric acid (76%) < [Cho]Cl/Malonic acid (81%) < [Cho]Cl/Lactic acid (98%). The same trend was observed for molecular weight (Mw): [Cho]Cl/Urea (75 kDa) < [Cho]Cl/Citric acid (84 kDa) < [Cho]Cl/Malonic acid (96 kDa) < [Cho]Cl/Lactic acid (125 kDa).”
- Page 3: Another important difference is that pulping results in chitin deacetylation (%DDA) to a significant extent (11%), whereas with the IL extraction %DDA nearly ideal (2(1))%.54 What does (2(1))% mean?
Response: The sentence was modified for clarification and consistency.
Revision (Lines 146-148): “Another important difference is that pulping results in chitin deacetylation (%DDA) to a significant extent (711%), whereas with the IL extraction %DDA nearly ideal (2 ± 1%).”
- The resolution of Figure 1 should be enhanced. In addition, I suggest that the authors should be further increase the number of pictures in main text for the clearer exhibition for readers.
Response: We appreciate the comment from the reviewer. We will work with the editorial office to make sure the quality of the figure is good for its publication. As of the number of figures, we respectfully disagree with the reviewer, since we do not see the need for additional schematics.
- From author's point view, what are the main problems to limit the use of DESs in industrial applications such as recovery of beneficial substances from biomass-based waste?
Response: Although this question needs to be answered separately for each DES-based process, we have included in the Conclusions section some general points, such as toxicity, recovery, and reusability that are still to be considered when assessing the feasibility of a process.
Reviewer 2 Report
This mini-review gives nice overview for DESs in the biotechnology. Some minor changes are required:
1) A comprehensive picture describing processes should be prepared and inserted.
2) The design of the table should be improved
3) There many typos and mistakes. For example on Page 8 Na2S and NaOH remained with strikethrough font:
The Kraft process employs a hot mixture of water, sodium hydroxide, (NaOH), and sodium sulfide (Na2S), and requires lots of energy to reduce sulfides-containing black liquor waste emissions. Handling this waste poses hazards to the environment.
Author Response
- A comprehensive picture describing processes should be prepared and inserted.
Response: Since we have described so many different processes, we’d respectfully disagree that a comprehensive picture would help the reader.
- The design of the table should be improved
Response: We will work closely with the journal’s editorial office to make sure that the table is presented in an accessible mode for the reader.
- There many typos and mistakes. For example on Page 8 Na2S and NaOH remained with strikethrough font:
Response: We appreciate the observation from the reviewer. We have corrected these and other typos present in the previous version of the manuscript.